# The Cardiovascular Disease (CVD) Risk Continuum from Prenatal Life to Adulthood: A Literature Review

**DOI:** 10.3390/ijerph19148282

**Published:** 2022-07-07

**Authors:** Maria Felicia Faienza, Flavia Urbano, Giuseppe Lassandro, Federica Valente, Gabriele D’Amato, Piero Portincasa, Paola Giordano

**Affiliations:** 1Department of Biomedical Sciences and Human Oncology, Pediatric Unit, University of Bari “A. Moro”, 70121 Bari, Italy; 2Giovanni XXIII Pediatric Hospital, 70126 Bari, Italy; flaviaurbano84@gmail.com (F.U.); giuseppelassandro@live.com (G.L.); 3Department of Cardiology, Erasme University Hospital, Université Libre de Bruxelles, 1050 Brussels, Belgium; federica.valente@ulb.be; 4Neonatal Intensive Care Unit, Di Venere Hospital, 70131 Bari, Italy; gab59it@yahoo.it; 5Clinica Medica “A. Murri”, Department of Biomedical Sciences and Human Oncology, University of Bari “A. Moro”, 70121 Bari, Italy; piero.portincasa@uniba.it; 6Department of Interdisciplinary Medicine, Pediatric Unit, University of Bari “A. Moro”, 70121 Bari, Italy; paola.giordano@uniba.it

**Keywords:** cardiovascular diseases (CVDs), risk factors, obesity, metabolic syndrome, prenatal, epigenetic, childhood, adulthood

## Abstract

The risk of developing cardiovascular diseases (CVDs) arises from the interaction of prenatal factors; epigenetic regulation; neonatal factors; and factors that affect childhood and adolescence, such as early adiposity rebound (AR) and social and environmental influences. Thus, CVD risk varies between the group of low-risk metabolically healthy normal-weight subjects (MHNW); the intermediate-risk group, which includes metabolically healthy obese (MHO) and metabolically unhealthy normal-weight subjects (MUHNW); and the high-risk group of metabolically unhealthy obese (MUHO) subjects. In this continuum, several risk factors come into play and contribute to endothelial damage, vascular and myocardial remodeling, and atherosclerotic processes. These pathologies can occur both in prenatal life and in early childhood and contribute to significantly increasing CVD risk in young adults over time. Early intervention in the pediatric MUHO population to reduce the CVD risk during adulthood remains a challenge. In this review, we focus on CVD risk factors arising at different stages of life by performing a search of the recent literature. It is urgent to focus on preventive or early therapeutic strategies to stop this disturbing negative metabolic trend, which manifests as a continuum from prenatal life to adulthood.

## 1. Introduction

Obesity is currently one of the most important public health problems in the United States and many other countries [1,2]. With increasing obesity prevalence, the incidence of several other associated comorbidities has also increased. This trend represents the enormous burden of obesity-related diseases worldwide [2,3]. It is therefore imperative that healthcare providers identify children at risk of overweight and obesity at any age, including the prenatal period. Notably, obesity during adolescence increases the risk of cardiovascular diseases (CVDs) and premature death during adulthood. This link is independent of obesity during adulthood [4,5,6,7,8].

CVD risk results from the interaction of prenatal, childhood, and adulthood risk factors. Prenatal risk factors include maternal body mass index (BMI), weight gain during pregnancy, maternal nutrition, fetal growth restriction, and epigenetic regulation; childhood risk factors include early adiposity rebound, obesity and metabolic syndrome (MS); and adulthood risk factors include endothelial dysfunction and early atherogenesis [9] (Figure 1). The National Institute of Health (NIH) has classified CVD risk factors into two main subgroups, i.e., non-modifiable (including age, gender, race/ethnicity, family history, and socioeconomic status) and modifiable. This latter group can be further divided into cardiometabolic factors such as hypertension; diabetes; an abnormal lipid profile; and lifestyle factors, which include physical inactivity, diet, and obesity [10]. The risk of developing cardiovascular disease manifests as a continuum that gradually increases from the low-risk metabolically healthy normal-weight (MHNW) group, through the intermediate risk group, including metabolically healthy obese (MHO) and metabolically unhealthy normal-weight (MUHNW) subjects, to the group of high-risk metabolically unhealthy obese (MUHO) individuals [11]. Several risk factors participate in this continuum, leading to endothelial damage, vascular and myocardial remodeling, and atherosclerotic processes. These changes may start in prenatal life or early childhood and over time significantly increase the CVD risk in young adults. Whether early intervention in the MUHO pediatric population can reduce CVD risk in adulthood remains to be determined.

Table 1 lists the metabolic and CV comorbidities in obese children and adolescents. In this review, we focus on the CVD risk factors that arise throughout the different stages of life by performing a review of the recent literature.

## 2. Methods

In this review, we focused on the pathogenesis of the CVD risk continuum, considering studies on prenatal, childhood, and adulthood risk factors for CVDs.

### 2.1. Eligibility Criteria

Manuscripts considered eligible for this review included: i. original published articles and ii. observational or experimental studies.

### 2.2. Information Sources and Search Strategy

The following key words were searched in PubMed and EMBASE: “cardiovascular diseases (CVDs)” AND “risk factors” AND “prenatal” AND “epigenetic” AND “obesity” AND “metabolic syndrome” AND “childhood” AND “adulthood”. The search was limited to the period from January 2005 to May 2022.

### 2.3. Study Selection

Articles were reviewed with regard to four main topics, i.e., prenatal, epigenetic, and childhood CVD risk factors and the concept of a continuum from childhood to adulthood.

### 2.4. Data Collection Process and Data Items

For each study that was considered eligible, the following data were collected: number and age of subjects, presence or lack of a comparison group, clinical parameters, and biochemical and instrumental markers of CVDs.

## 3. Prenatal CVD Risk Factors and Epigenetic Regulation

### 3.1. Prenatal Risk Factors

The most important prenatal CVD risk factor is fetal growth restriction (FGR), which affects 7–10% of pregnancies and is defined as a failure to achieve the genetic growth potential [12]. Epidemiologic studies have demonstrated that a low birth weight (LBW) is associated with an increased risk of coronary artery disease (CAD), also called coronary heart disease (CHD), and stroke [13,14,15]. Furthermore, children born small for gestational age (SGA) have an increased risk of developing permanent metabolic changes that lead to increased CVD risk [16]. This phenomenon is referred to as “fetal programming” [17], a condition which influences the development of CVDs through two main pathways: metabolic programming and cardiovascular reprogramming.

Metabolic programming is a nutritional intrauterine and/or early postnatal event that occurs during a critical period of development and has lasting or lifelong consequences. According to the Developmental Origin of Health and Disease (DOHaD) hypothesis, metabolic programming is caused by epigenetic modifications of non-imprinted genes induced by the intrauterine environment [18,19,20]. Undernutrition, macronutrient excess, and/or stress during intrauterine life trigger adaptive responses, leading to insulin resistance and increased risk of CVDs and metabolic diseases later in life.

Through cardiovascular reprogramming, FGR has a lifelong impact: during fetal and early postnatal life, it is responsible for heart remodeling, increased intima–media thickness (IMT), an abnormal atherogenic lipid profile, and the loss of nephrons [21,22]. During fetal life, the sustained restriction of nutrients and oxygen due to placental insufficiency has two direct effects on fetal CV development: the disruption of myocardial fibers and increased placental resistance and chronic volume/pressure overload. Consequently, the myocardium develops changes in its macro- and microstructure and function, which is defined as cardiac remodeling, to maintain the ventricular output [23]. This may occur in one ventricle (“elongated” phenotype, where a globular right ventricle pushes the septum and elongates the left ventricle) or both ventricles (“globular” phenotype). In more severe and/or prolonged cases, increased sphericity may not be enough, and so hypertrophy develops to increase contractility and decrease local wall stress. Generally, early-onset FGR is more strongly associated with a hypertrophic response, whereas late-onset FGR usually develops a globular or elongated morphology.

In childhood, FGR causes increased blood pressure, which persists in young adults. Preterm birth or the SGA condition are associated with smaller kidneys, a lower nephron number, an abnormal nephron morphology, and a reduced glomerular filtration rate, which are responsible for the development of hypertension and impaired renal function later in life [24,25,26]. Furthermore, preterm birth entails the prenatal and postnatal exposure to intensive treatments such as steroids, nephrotoxic drugs, and infections, potentially impacting nephron development. Moreover, hormonal factors induced by fetal growth restriction, such as high levels of insulin-like growth factor-1 (IGF-1), as well as protein restriction in the maternal diet, are probably involved in the onset of high blood pressure in SGA newborns or those with FGR.

Other factors, such as maternal obesity, hypertension, endothelial dysfunction, insulin resistance, and diabetes may influence fetal metabolic programming and cardiovascular reprogramming, leading to the development of CVDs [27]. However, the role of these factors in CVD programming is limited compared to FGR.

### 3.2. Epigenetic Regulation

The intrauterine environment plays an important role in the complex interplay between genes and the epigenetic mechanisms that regulate their expression. Epigenetic processes are tightly regulated during embryonic and fetal growth and play important roles in the normal development of organs, including the heart. It is likely that changes in the intrauterine environment, such as hypoxemia and the dysregulation of maternal nutrient intake, have an impact on the epigenome during pregnancy, potentially generating lifelong consequences [28].

The control of fetal programming is mediated by three epigenetic pathways: (i) DNA methylation, (ii) histone modifications, and (iii) the expression of microRNA (miRNA) [29,30]. These miRNAs, small non-coding RNA molecules containing about 22 nucleotides, are involved in insulin signaling, glucose transport, insulin resistance, and cholesterol and lipid metabolism. In a pilot study on circulating miRNA signatures in children with obesity born small for gestational age (SGA) and appropriate for gestational age (AGA), a specific profile of circulating miRNAs was observed in both groups of obese children compared with children of a normal weight [31].

In addition, maternal metabolic disorders, such as gestational diabetes and hypertension, and maternal stress during pregnancy can dysregulate the expression of key miRNAs involved in the development of a healthy heart [32]. Although the exact role of miRNAs in the epigenetic regulation of cardiac development has not been fully elucidated, in vivo studies have demonstrated the importance of certain miRNAs, such as microRNA-1 (miR-1) and microRNA-133a (miR133a), in ventricular septal defects or chamber dilatation [33,34]. A specific group of miRNAs are expressed exclusively in the placenta and are secreted into the fetal circulation [35]. The expression of these miRNAs is altered in the placenta during SGA pregnancies and in mothers exposed to environmental pollutants [35,36]. Although the role of these placental miRNAs is not fully understood, the hypothesis is that placental insufficiency can alter miRNA expression, with an impact on fetal heart development.

## 4. Childhood CVD Risk Factors

### 4.1. Obesity

Obesity is the most important CVD risk factor, and it is strongly associated with cardiometabolic comorbidities such as hypertension, dyslipidemia, hyperinsulinemia, type 2 diabetes, and non-alcoholic fatty liver disease (NAFLD) [37]. An increase in the BMI value before 6 years of age, which corresponds to the “adiposity rebound” (AR) stage, represents a risk factor for the development of obesity. In addition, an excess of visceral fat, a marker of central obesity that is measured as the waist circumference (WC) and waist circumference/height (WC/H) ratio, is a better predictor of CVD risk than BMI in children, and it may help to define the MUHO population [38,39]. Obesity is characterized by chronic low-grade systemic inflammation due to the increased secretion of proinflammatory cytokines by adipocytes and the infiltration of macrophages into the adipose tissue [40,41,42]. These cytokines trigger local effects in the endothelium by stimulating the production of vascular cell adhesion molecule-1 (VCAM-1) and intercellular adhesion molecule-1 (ICAM-1) and increasing the vascular permeability. Among the adipocytokines, leptin activates endothelial cells and promotes the infiltration of macrophages into the adipose tissue, while resistin induces the expression of VCAM-1 and ICAM-1 in vascular endothelial cells and promotes the secretion of proinflammatory cytokines such as tumor necrosis factor alpha (TNF-α), interleukin-6 (IL-6), and interleukin-12 (IL-12) [43,44].

Endothelial dysfunction is a key factor in the pathogenesis of atherosclerosis. Damage to the endothelium induced by the overexpression of proinflammatory cytokines changes the balance between vasoconstriction and vasodilation and activates several prothrombotic and proatherogenic processes that promote atherosclerosis; these include increased endothelial permeability, platelet aggregation, leucocyte adhesion, oxidative stress, and cytokine production [45,46]. The decreased production or activity of nitric oxide (NO), resulting in impaired vasodilation, may be one of the earliest signs of atherosclerosis in obesity [47]. Obesity in children has also been associated with decreased arterial elasticity in adulthood [48].

### 4.2. Components of MS

MS is defined by the presence of at least three out of five indicators: abdominal obesity, hypertension, high triglycerides levels, low high-density lipoprotein cholesterol (HDL-C), and impaired glucose metabolism, with specific cut-off values for each feature in children and adults [49].

Insulin resistance has a key role in the pathogenesis of MS, and it explains the association between obesity and vascular dysfunction [50,51,52]. In the presence of insulin resistance, the NO synthesis stimulated by insulin is impaired, and the compensatory hyperinsulinemia may activate the mitogen-activated protein kinase (MAPK) pathway, resulting in a vasoconstriction enhancement, inflammation, increased sodium, water retention, and, finally, elevated blood pressure. In addition, insulin resistance in endothelial cells causes an increase in prothrombotic factors, proinflammatory markers, and reactive oxygen species (ROS), which lead to a rise in the intracellular levels of ICAM-1 and VCAM-1. The detrimental effects of hyperglycemia on cardiomyocytes can be explained by a phenomenon called “hyperglycemic memory”, which refers to the long-term persistence of hyperglycemic stress even after blood glucose normalization. Hyperglycemia also increases proinflammatory and procoagulant factor expression and impairs NO release, leading to endothelial dysfunction [53]. Another component of MS is hyperuricemia, which predisposes to CV damage [54]. Two distinct mechanisms have been proposed to explain this link. First, hyperuricemia can induce endothelial dysfunction via insulin-stimulated NO-induced vasodilation. The second hypothesis considers the role of the xanthine produced in the ROS reaction, which contributes to oxidative and inflammatory alterations in adipocytes [55]. Among the adipokines, adiponectin plays a cardio-protective role by inhibiting TNF-mediated monocyte adhesion, the formation of foam cells, and smooth muscle cell proliferation and by promoting blood vessel growth and endothelial NO production [56,57].

Lipids and lipoproteins play an important role in the development and consequences of MS. Biomarkers such as apolipoprotein A-1 (apo A1) and apolipoprotein-B (apo B) have been proposed as predictors of atherogenicity and CVD risk [58]. The loss of the suppressive effects of insulin on lipolysis in adipocytes increases free fatty acid flux to the liver, which stimulates the assembly and secretion of very-low-density lipoprotein (VLDL), resulting in hypertriglyceridemia. The excess of lipids in the cardiomyocytes channeled into non-oxidative pathways results in the accumulation of toxic lipid species (lipotoxicity), which alters cellular signaling and cardiac structure. Disruptions in several cellular signaling pathways, such as during mitochondrial dysfunction and endoplasmic reticulum stress, have been associated with lipotoxicity. Mediators such as ROS, NO, ceramide, phosphatidylinositol-3-kinase (PI3K), diacylglycerol (DAG), ligands of peroxisome proliferator activated receptors (PPARs), and leptin have been proposed to promote these lipotoxic effects and enhance the rate of apoptosis [59].

## 5. CVD Continuum from Childhood to Adulthood

Longitudinal studies have found that being obese or overweight in early life may be associated with increased atherosclerosis and morbidity and mortality from CVDs. Obese children and adolescents are five times more likely to become obese adults. About 55% of obese preschool children continue to be obese in adolescence, and about 80% of obese adolescents will continue to be obese in adulthood [60]. The most important factors influencing the persistence of obesity in adulthood are the age at the onset of obesity, the severity of obesity, and the presence of parental obesity [61]. Adult obesity carries an increased risk of CVDs. Juonala et al. demonstrated that subjects who had been overweight or obese in youth had significantly higher carotid IMT values in adulthood compared with subjects who had been lean in youth, while subjects who had been obese in youth but were non-obese as adults had IMT values comparable with subjects who had remained consistently non-obese [62]. In obese children, endothelial dysfunction is related to the severity of obesity and the degree of insulin resistance, and it contributes to early atherogenesis during childhood, because endothelial cells are important in the regulation of vasomotion, thrombosis, and inflammation [40]. An epidemiologic study on the early natural history of CVDs in children and young adults, the Bogalusa Heart Study (BHS), demonstrated a correlation between clinical risk factors in early life and anatomic changes in the aorta, coronary vessels, and cardiac and renal systems related to atherosclerosis and hypertension [63]. Although clinical CVDs occur in later life, indicators of atherosclerosis, hypertension, and diabetes mellitus are clearly present in childhood. According to the BHS, atherosclerosis is evident early in life, and the degree of atherosclerosis in children is associated with the presence of cardiometabolic risk factors in childhood.

High blood pressure, dyslipidemia, impaired glucose metabolism, and systemic inflammation in children are the main risk factors associated with the incidence of premature atherosclerosis [64,65]. If not adequately treated, these factors may contribute to an increased risk of adverse CV events in adulthood [66].

In particular, the stigmata of obesity in children and adolescents are associated with cardiovascular changes linked to increased adulthood CVD risk, including hypertension and dyslipidemia, which are components of MS. A minority of children will remain metabolically healthy in adulthood [67]. Thus, a more accurate and earlier understanding of this risk stratification could improve the assessment and management of CVD risk factors. Children and adolescents with overweight and obesity show an increased risk of hypertension, and this risk increases with the severity of obesity [68,69]. One study found a 4% prevalence of hypertension in children with moderate obesity and a 9% prevalence in those with severe obesity [70]. In another study, the risk of hypertension was twofold higher and fourfold higher in children with mild and severe obesity, respectively, compared with children of a normal weight [71]. Notably, the presence of hypertension during childhood predicts hypertension and MS during adulthood [72]. However, the risk resolves if the individual loses weight by adulthood [73,74].

As for adults, obese children show elevated serum concentrations of LDL-C and triglycerides and decreased HDL-C, with the risk of dyslipidemia increasing with the severity of obesity [75,76]. Obese children are at risk of deranged cardiac structure and function, including increased left ventricular mass, developing in both hypertensive and non-hypertensive children with obesity [77,78]. Additional abnormalities include an increased left ventricular and left atrial diameter, systolic and diastolic dysfunction, and increased epicardial fat. Worryingly, if intima–media thickening appears in obese individuals during childhood and adolescence, it will persist even if individuals lose weight by adulthood [74].

## 6. Conclusions

The risk of developing CVDs begins before and during pregnancy and is maintained at different stages of life, in the presence of other predisposing factors. Maternal feeding during the prenatal period can affect the offspring by increasing the susceptibility to cardiometabolic diseases. In particular, maternal obesity can lead to changes in DNA methylation at birth, which persist throughout different stages of life. The association between a low birth weight and cardiovascular diseases and diabetes confirms that malnutrition during the early stages of development permanently “programs” organ structure and metabolism. Literature data indicate epigenetic mechanisms as possible mediators for the development of cardiometabolic diseases. Obesity and its comorbidities, especially when early-onset, can predispose to the development of endothelial damage and early atherosclerosis.

Prevention through maintaining an adequate lifestyle and proper nutrition involving bioactive food compounds could neutralize the epigenetic abnormalities induced by various environmental factors.

## Figures and Tables

**Figure 1 ijerph-19-08282-f001:**
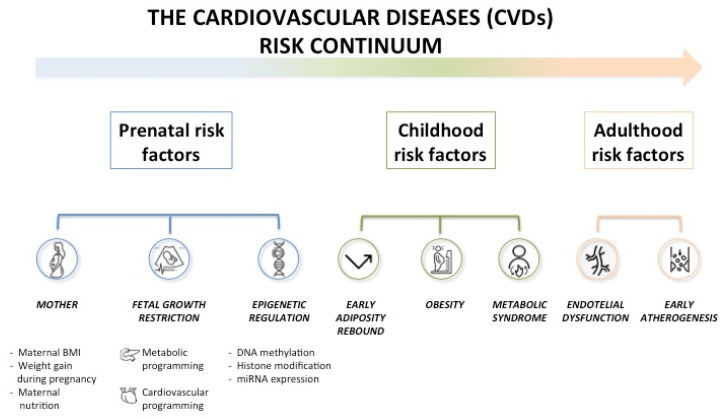
The cardiovascular disease (CVD) risk continuum.

**Table 1 ijerph-19-08282-t001:** Metabolic alterations and cardiovascular comorbidities in obese children and adolescents.

**Metabolic Alterations**
a. Insulin resistance b. Prediabetes (impaired fasting glucose/impaired glucose tolerance) c. Type 2 diabetes d. Dyslipidemia e. Metabolic syndrome
**Cardiovascular Comorbidities**
a. Hypertension b. Endothelial dysfunction c. Abnormal cardiac structure and function d. Premature atherosclerotic cardiovascular disease

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
