# Peer review of "The Cardiovascular Disease (CVD) Risk Continuum from Prenatal Life to Adulthood: A Literature Review"

_ijerph, 2022, doi:10.3390/ijerph19148282_

Round 1
Reviewer 1 Report
Please check attached the file

Author Response
Responses to Reviewer 1
1.The title is suggested to be adjusted to the content of the manuscript.
Answer: we adjusted the title according to the reviewer suggestion.
2-9. Abstract:
Answer: we corrected according to the reviewer suggestion.
- Abstract:
Answer: we corrected as suggested
- Introduction: add the figure 1 in the manuscript
Answer: we added the figure 1 in the manuscript.
12.Table 1 (P2, L66) Please be redrawn the Table 1. Please give the readers
more pieces of information
Answer: we have redrawn the table 1.
- What does the meaning of timely literature search?
Answer: the meaning is “punctually”
- Please be added their years, for examples: from ….to May 2022
Answer: we added the years
- Please be rewritten the Method systematically and in detail. Please give readers more information.
Answer: we have rewritten the method sistematically.
16-19: Correct CVDs
Answer: we corrected as suggested
20-54: Correct the words/sentences
Answer: we corrected as suggested
55: References
Answer: we corrected references according journal guidelines
- Please be rewritten theses sentences.
Answer: we have rewritten the sentences.
- Please be rewritten the following paragraphs because they are too long, namely: 1. P4, L150-L179. 2. P5, L195-L223.
Answer: we have rewritten the paragraphs as suggested.

Reviewer 2 Report
This work is a narrative review. Although important, it is not suitable for this journal. Please consider reformulating all the work in order to perform a systematic review, if possible with a metanalysis, if none has been performed before (search pubmed, embase, central, prospero and/or other databases). If none has been published or registered, then your work will be suitable to publish in this journal.
Author Response
This work is a narrative review. Although important, it is not suitable for this journal. Please consider reformulating all the work in order to perform a systematic review, if possible with a metanalysis, if none has been performed before (search pubmed, embase, central, prospero and/or other databases). If none has been published or registered, then your work will be suitable to publish in this journal.
Answer:
We thank the reviewer for the suggestion. Our aim was to seek updated literature data explaining predisposing factors for the development of cardiovascular disease starting from prenatal life to adulthood. It was not our aim to carry out a metanalysis which considers a series of mathematical-statistical methods that allow to aggregate and combine data of different studies with reduced sample size or with discordant results, conducted on the same topic.

Reviewer 3 Report
This review article is well searched and well presented. Please find attached my comments. Thanks.

Author Response
Reviewer 3
Thanks for the opportunity to review this important paper. As a review article, it has done well by
searching wide range of relevant articles. The flow of ideas is logical and seamless.
Please see below specific comments for your consideration.
Line Comment
186 oxyde change to oxide
204 ROS abbreviation appearing for the first time, please define it first.
232/233 Mediators
259 Full stop
284 obesity change to obese
Answer: we corrected the mistakes as suggested.

Reviewer 4 Report
Dear Editor,
I really appreciate the opportunity to review the manuscript ijerph-1758736 entitled:
"The Cardiovascular Risk Continuum from Prenatal Life to Adulthood"
I commend the authors for describing this critical and timely issue. The paper is interesting and well-written; however, I would like to highlight some issues that merit revision:
The methodology is absolutely adequate as well as the description of the results, also good technical writing of the article. Although it is not an article about treatment, for completeness it would be good to specify, at least in a few lines in point 5. "5. CV continuum from childhood to adulthood," whether or not there is, whether it is possible or is being studied in any way, a contrast to fetal programming; it is well known that studies are at the less than initial stages, but it should be specified in the manuscript. It should also be emphasized more emphatically in the conclusions how early treatment at least at the symptom level can influence improving the final outcome.
Author Response
Dear Editor,
I really appreciate the opportunity to review the manuscript ijerph-1758736 entitled:
"The Cardiovascular Risk Continuum from Prenatal Life to Adulthood"
I commend the authors for describing this critical and timely issue. The paper is interesting and well-written; however, I would like to highlight some issues that merit revision:
The methodology is absolutely adequate as well as the description of the results, also good technical writing of the article. Although it is not an article about treatment, for completeness it would be good to specify, at least in a few lines in point 5. "5. CV continuum from childhood to adulthood," whether or not there is, whether it is possible or is being studied in any way, a contrast to fetal programming; it is well known that studies are at the less than initial stages, but it should be specified in the manuscript. It should also be emphasized more emphatically in the conclusions how early treatment at least at the symptom level can influence improving the final outcome.
Answer:
We really thanks to the referee for his/her valuable suggestion. We have rewritten the conclusions pointing out on the possible methos to contrast fetal programming.

Round 2
Reviewer 1 Report
Please check the attached file

Author Response
Reviewer Comments
1 Introduction Please be rewritten these sentences based on Figure 1 and make them
systematically: CVDs risk results from the interaction of prenatal factors including maternal body
mass index (BMI), weight gain during pregnancy, maternal nutrition, as well as epigenetic
regulation, birth weight, early adiposity rebound, and social and environmental influences
occurring during childhood and adolescence [9]
Answer: we have rewritten the phrases according with figure 1.
2 Figure 1 Please be given a legend for Figure 1.
Answer: we added the figure 1 legend.
3 Figure 1 Please be redrawn Figure 1. And then, please be made it eyes-catching and fully pieces
of information.
Answer: we adjusted the figure 1 according with the text.
4 The legend of Table 1 CV comorbidities >>> cardiovascular comorbidities
Answer: we adjusted the table 1 legend.
5 Table 1 CARDIOVASCULAR >>> CARDIOVASCULAR COMORBIDITIES
Answer: we changed as suggested.
6 Conclusion Please be rewritten the conclusions because of look like literature. And then, please
be deleted their numbers, such as numbers [80], [81], and [82] in their sentences.
Answer: we have rewritten the text.
7 Conclusion Please be deleted the references of number [80], [81], and [82].
Answer: we deleted the references as suggested

Reviewer 2 Report
Section 2 – Methods
- Authors should define their PICO/PICOTS and afterward find synonyms for each one of the terms identified for each item of the PICO/PICOTS. Then, it is possible to define the “search sentence” to search databases systematically
- Please justify the dates considered. When (date and time) was the search performed? This is one of the items that PRIISMA states to register (and authors mentioned that followed the PRISMA),
- Please define inclusion/exclusion criteria for papers, according to the PRISMA statement. Here, authors can define the range period (but justifying the start date, if any), language, etc. and any filters used
- Please define how many authors (and who) read the papers to decide which papers should be included or not, how were disagreements handled and how many rounds were performed (and what was evaluated at each round)
- Have you registered the protocol. If so, where? (PROSPERO?)
- Authors do not present any flow chart as mandatory when using the PRISMA statement or in any systematic review, as this manuscript is defined to be at its title
- Authors do not present any data extraction table as mandatory in any systematic review, as this manuscript is defined to be at its title
- Have you analysed the quality and bias of the papers included? If so, which tool have you considered? NOS? QUADAS? Other? Why?
- In a systematic review, there is always a qualitative analysis of the findings, which is not reflected in your manuscript (certainly that there exist a critical discussion and a conclusion).
To conclude, there is no evidence that the PRISMA statement was followed to obtain this review. A systematic review considers the scientific method and this is not visible in your manuscript as your work has not begun to be a systematic review.
Author Response
Answer: We really thank the referee for his/her remark.
We agree with you that we have probably not followed all the recent recommendations of
PRISMA 2020. However, we have recently published in this journal a systematic review
“Giordano P, Urbano F, Lassandro G, Faienza MF. Mechanisms of bone impairment in sickle cell
anemia. Int J Environ Res Public health. 2021 Feb 13; 18 (4): 1832. doi: 10.3390 / ijerph18041832.
following the same method used for this review.
If you think this is not a systematic review we can change the title with a "literature review"
